# Epigenetic Regulation in Lean Nonalcoholic Fatty Liver Disease

**DOI:** 10.3390/ijms241612864

**Published:** 2023-08-16

**Authors:** Ioanna Aggeletopoulou, Maria Kalafateli, Efthymios P. Tsounis, Christos Triantos

**Affiliations:** 1Division of Gastroenterology, Department of Internal Medicine, University Hospital of Patras, 26504 Patras, Greece; iaggel@hotmail.com (I.A.); makotsouno@gmail.com (E.P.T.); 2Department of Gastroenterology, General Hospital of Patras, 26332 Patras, Greece; mariakalaf@hotmail.com

**Keywords:** epigenetics, nonalcoholic fatty liver disease, lean NAFLD, nonobese NAFLD, epigenetic regulation, DNA methylation, histone modification, noncoding RNAs

## Abstract

Nonalcoholic fatty liver disease (NAFLD), the most prominent cause of chronic liver disease worldwide, is a rapidly growing epidemic. It consists of a wide range of liver diseases, from steatosis to nonalcoholic steatohepatitis, and predisposes patients to liver fibrosis, cirrhosis, and even hepatocellular carcinoma. NAFLD is strongly correlated with obesity; however, it has been extensively reported among lean/nonobese individuals in recent years. Although lean patients demonstrate a lower prevalence of diabetes mellitus, central obesity, dyslipidemia, hypertension, and metabolic syndrome, a percentage of these patients may develop steatohepatitis, advanced liver fibrosis, and cardiovascular disease, and have increased all-cause mortality. The pathophysiological mechanisms of lean NAFLD remain vague. Studies have reported that lean NAFLD demonstrates a close association with environmental factors, genetic predisposition, and epigenetic modifications. In this review, we aim to discuss and summarize the epigenetic mechanisms involved in lean NAFLD and to introduce the interaction between epigenetic patterns and genetic or non genetic factors. Several epigenetic mechanisms have been implicated in the regulation of lean NAFLD. These include DNA methylation, histone modifications, and noncoding-RNA-mediated gene regulation. Epigenetics is an area of special interest in the setting of lean NAFLD as it could provide new insights into the therapeutic options and noninvasive biomarkers that target this under-recognized and challenging disorder.

## 1. Introduction

Nonalcoholic fatty liver disease (NAFLD) is a common liver condition affecting approximately 25% of the global population (20–30% of the Western population [1,2] and up to 34.2% of obese children) [3]. It is defined as a chronic liver disease, characterized by steatosis in liver cells, in patients with no remarkable alcohol consumption and without other liver disorders [4]. NAFLD encompasses a wide range of liver diseases, from simple fat accumulation to more advanced stages such as nonalcoholic steatohepatitis (NASH), liver cirrhosis, and even hepatocellular carcinoma (HCC) [5]. Although NAFLD is typically asymptomatic, 25% of patients with NASH may progress to liver cirrhosis, and 10% may develop decompensated liver disease [6]. Currently, liver-related mortality constitutes the third leading cause of death in NAFLD individuals [7,8]; however, due to its rapidly increasing prevalence, NASH-related fibrosis is expected to become the primary cause of liver transplantation in the near future [9,10]. In parallel, the rising prevalence of NAFLD, due to its close relation to other comorbidities such as obesity, cardiovascular disorders, type 2 diabetes mellitus (T2DM), and other metabolic abnormalities, further increases the morbidity and mortality rates of this disorder [11,12].

The increasing occurrence of NAFLD is strongly related to the worldwide obesity crisis [13]; however, up to 10–20% of individuals diagnosed with NAFLD present with normal body mass index (BMI) [14,15]. This specific group is referred to as “lean NAFLD” or “nonobese NAFLD” [14,15,16]. The lean NAFLD phenotype was initially observed in Asian populations, but it can also occur in other ethnic groups and may indicate visceral obesity in the absence of systemic obesity [17]. As NAFLD is mainly associated with the Western definition of obesity and metabolic syndrome, it may go under-recognized or completely undetected in lean populations [18]. The global incidence of lean NAFLD is substantially rising [19]. Beyond Asians, lean NAFLD has also been documented in other populations, with an incidence of 8–20% [20]. The increasing incidence of NAFLD is reported by many studies, which either use the obesity definition as defined by the World Health Organization (WHO) or an ethnicity-based BMI cutoff [21]. Although the lean NAFLD phenotype typically presents with a less severe form of the disease, it may also exhibit a wide range of histopathological characteristics associated with NASH, including steatosis, hepatocyte ballooning, lobular inflammation, and liver fibrosis [22]. Furthermore, individuals with lean NAFLD encounter comparable complications and comorbidities to obese patients [23,24]. Recently, lean NAFLD has been categorized into two subtypes based on disease epidemiology, prognosis, and natural history [25]. Type 1 mainly occurs in patients with visceral adiposity and insulin resistance, whereas type 2 refers to a condition that affects patients with hepatic steatosis caused by monogenic diseases [25,26]. Most lean NAFLD patients fall into type 1, meeting the BMI criteria for being lean but exhibiting obesity based on waist circumference or other body composition measures [25,26]. The NAFLD pathophysiology in this subtype is likely similar to that seen in overweight and obese NAFLD individuals, as hepatic steatosis and lipotoxicity are promoted by excessive food consumption, particularly simple carbohydrates and a sedentary lifestyle [27]. On the other hand, lean NAFLD in the absence of visceral adiposity may be promoted by the occurrence of rare genetic variants [28,29].

This evidence suggests that NAFLD in both obese and lean individuals imposes a significant burden on society. Thus, the elucidation of molecular mechanisms underlying the disease pathogenesis is crucial, especially in lean NAFLD, which has been characterized as an under-recognized and challenging disorder. In this review, we will focus on the epigenetic modulation in lean NAFLD and discuss the recent progress regarding the role of epigenetics in this condition as well as the underlying mechanisms. In addition, we will further discuss the potential application of epigenetic modulators in clinical practice using them as biomarkers and novel therapeutic targets in lean NAFLD treatment.

## 2. NAFLD Pathogenesis—The Role of Epigenetics

The pathogenesis of NAFLD is complex and involves multiple elements, including genetic, metabolic, and environmental factors. Although the exact mechanisms are not fully understood, several key factors have been closely implicated in the development and progression of NAFLD. Growing evidence highlights the critical role of epigenetic regulation in the development and progression of NAFLD [25,30,31,32,33].

Epigenetics involves heritable modifications to the structure and biochemistry of chromatin, without altering the DNA sequence [34,35]. The epigenetic mechanisms modulate diverse physiological and pathological processes via the regulation of gene expression through alterations in epigenetic code accessibility within the chromatin [36]. Epigenetic modulation can be performed at various levels; however, three major epigenetic codes that have been extensively studied are DNA methylation, histone modifications, and noncoding RNA (ncRNA) [37,38,39]. Diverse biological functions and phenotype–environment interplay in response to various stressor stimuli are substantially influenced by epigenetic pathways promoting phenotypic plasticity [40,41]. Epigenetic regulatory mechanisms are reversible, as lifestyle and environmental changes can determine epigenetic patterns during life and have the potential for dynamic modulation; thus, epigenetic-associated alterations to genes and proteins may serve as future therapeutic strategies in the clinical setting [42,43].

These modifications can alter the expression of the genes involved in lipid metabolism, inflammation, and oxidative stress, all of which contribute to the development of NAFLD. Cytosine and histone modifications, as well as alterations in the localization of nucleosomes occurring at the molecular level, are potential drivers of epigenetic regulatory mechanisms [44]. Growing evidence suggests that lean NAFLD progression is substantially influenced by multiple epigenetic mechanisms [32,45]. Among these epigenetic alterations, the modifications of the amino-terminal ends of histones are particularly important for maintaining the chromatin structure and regulating gene expression [46]. Additionally, abnormal DNA methylation serves as an initial event in the development of cancer in patients with NAFLD [47]. Furthermore, the assessment of circulating microRNA (miRNA) profiles holds promise as a noninvasive approach to evaluate and monitor the severity of liver disease [33,48].

## 3. DNA Methylation

DNA methylation, a key player in epigenetic transcriptional silencing, is a heritable epigenetic process that involves the covalent transfer of a methyl group (CH_3_) to the C5 position of the cytosine ring of DNA to form 5-methylcytosine; this process is catalyzed by DNA methyltransferases (DNMTs) [49] (Figure 1).

DNA methylation can cooperatively regulate the chromatin state through the interactions among DNMTs and components of the chromatin machinery [49]. Through the deposition of histone marks at methylation sites, DNA methylation can be stably inherited without altering the DNA sequences [50]. The identification of aberrant DNA methylation patterns may potentially supply novel treatment targets and biological tools for the diagnosis and/or prognosis of NAFLD [51].

Recent data have shown that blood DNA methylation markers may differentiate NAFLD patients with nonsignificant liver fibrosis from those with significant fibrosis [52]. Several genes have been identified as potential blood methylation biomarkers for the diagnosis of liver fibrosis in NAFLD, including CISTR, IFT140, and RGS14 [52]. Such genes that include differentially methylated probes in their DNA could potentially provide some evidence on the role of DNA methylation in the liver fibrosis progression of NAFLD patients.

Phosphatidylethanolamine N-methyltransferase (PEMT) is a 22.3 kDa transmembrane protein, which is responsible for the transfer of methyl groups to cytosine and the catalyzation of phosphatidylethanolamine (PE) to phosphatidylcholine (PC) [53] (Figure 1). Although liver is the tissue with major PEMT activity, low levels of PEMT in adipocytes have been associated with lipid droplet formation [54]. Experimental animal studies have shown that PEMT activity is necessary for the maintenance of the hepatic membranes’ integrity and PC production when dietary choline availability is limited [55,56]. PEMT is also required for the proper secretion of very low-density lipoproteins [57].

Whole-exome sequencing was performed to investigate causative alterations in the common DNA nucleotide sequences related to disrupted liver fatty acid metabolism in patients with lean NAFLD [58]. The variants in PEMT and oxysterol-binding protein-related protein10 (OSBPL10) genes, which are commonly related to dietary choline intake and cholesterol metabolic modulation, respectively, were identified as potential biomarkers and were then processed for further validation [58]. Although no association was demonstrated between the variant rs2290532-OSBPL10A and the risk of lean NAFLD, a significant association was observed between the variant rs7946-PEMT and the susceptibility to disease [58] (Figure 1). A previous study reported that the variant Val175Met of the PEMT gene could be a prognostic biomarker for susceptibility to NASH, as this genetic variant was more frequently demonstrated in lean NASH patients [59] (Figure 1).

PEMT knockout (KO) mice have been used as genetic models for lean NAFLD [56]. Studies have shown that, although PEMT KO mice demonstrated protection against diet-induced obesity and insulin resistance [60], they were more susceptible to diet-induced fatty liver and steatohepatitis [60]. On the other hand, a choline-deficient diet was unable to substitute the de novo synthesis of intracellular PC in the liver [55,60].

Experimental diet-induced lean NAFLD models include those of choline-deficient amino acid-defined (CDAA) diet [61], methionine–choline-deficient (MCD) diet [62], and high-fat–high-fructose diet or combined diets [63]. In the MCD diet, the absence of methionine and choline resulted in impaired PC and very low-density lipoprotein production [64]. This resulted in decreased triglyceride clearance and the promotion of lipid accumulation in the liver [65]. Mice fed with an MCD diet typically experienced a weight loss of around 40% [66]. Compared with these mice, PEMT KO mice did not experience significant weight loss and displayed a better phenotypic resemblance to lean NAFLD mice [67]. PEMT KO mice in a high-fat diet exhibited several phenotypic similarities with MCD-fed mice [68]; however, MCD-fed mice displayed decreased liver weight relative to body weight [60].

Differentially methylated regions (DMRs) have been implicated in the development of hepatocarcinogenesis in lean (choline-deficient model) versus obese (choline-supplemented model) mice with NASH [69]. Obese mice displayed a greater number of DMRs during the progression from NASH to HCC, compared with nonobese mice [69]. In lean mice with NASH-HCC, variations in methylation were observed in the genes associated with cancer progression and prognosis, including HCC-related genes such as CHCHD2, FSCN1, and ZDHHC12, as well as the genes involved in lipid metabolism such as PNPLA6 and LDLRAP1 [69]. Conversely, in obese mice with NASH-HCC, methylation differences were found in the genes already known to be linked with HCC, such as RNF217, GJA8, PTPRE, PSAPL1, and LRRC8D [69]. Hypomethylated DMRs in obese NASH-HCC mice were enriched in the genes related to Wnt signaling pathways, suggesting that HCC progression in obese mice is potentially influenced by the hypomethylation of the genes associated with the Wnt signaling pathway, whereas in lean mice, it may be affected by alternative signaling pathways, such as those related to lipid metabolism [69].

### DNA Methylation Differences between Obese and Lean NAFLD

DNA methylation can influence gene expression patterns and may contribute to various physiological and pathological processes, including obesity and NAFLD. DNA methylation changes have been observed in the genes related to lipid metabolism, insulin sensitivity, and inflammation in obese NAFLD [51,70]. Altered methylation patterns in these genes may contribute to the development and progression of NAFLD in obese individuals [51,70]. The epigenetic regulation of adipogenesis has also been described in obese NAFLD. DNA methylation patterns in the genes involved in adipocyte differentiation and adipogenesis might be altered in such patients. These changes can affect how adipocytes store and release fatty acids, which can influence the overall hepatic lipid [70,71]. In parallel, epigenetic modifications in obesity-related inflammation can contribute to the progression of obese NAFLD to more severe stages [72]. Lastly, alterations in DNA methylation have been reported in pathways regulating insulin sensitivity and glucose metabolism, as insulin resistance is a common feature of obesity and NAFLD [73].

## 4. Histone Modification

The packaging of chromosomal DNA within nuclei is facilitated by positively-charged proteins, called histones, which are tightly bound to negatively charged DNA and assemble into nucleosome complexes [74,75]. Histones serve as the primary protein components of chromatin and are crucial for gene regulation. There are five major families of histone proteins, namely H1, H2A, H2B, H3, and H4 [76] (Figure 2).

The packaging of nuclear chromatin is modulated through various mechanisms. One of these mechanisms is the replacement of canonical histones with the histone variants that are incorporated into chromatin [77]. These histone variants play an essential role, highlighting their importance in epigenetic regulation [78,79,80]. Among these variants, macroH2A1 is a variant of the H2A family and exists in two alternatively exon-spliced isoforms: macroH2A1.1 and macroH2A1.2 [80,81]. These isoforms are critical for the regulation of cell proliferation and plasticity [80,81,82]. There is a great number of distinct histone post-translational modifications; these modifications not only play an essential role in regulating the chromatin structure but also actively recruit remodeling enzymes that use ATP-derived energy to reposition nucleosomes [83]. The recruitment of various proteins with specific enzymatic functions is now a well-established concept for how modifications exert their functional activities [83].

### Histone Modification in Lean MAFLD Patients

In 2020, a group of international experts suggested changing the term NAFLD to metabolic-associated fatty liver disease (MAFLD) to address some of the limitations and challenges associated with the NAFLD terminology [84]. MAFLD places a stronger emphasis on the metabolic aspects of the disease. The diagnostic criteria for MAFLD are broader and include not only the presence of liver fat but also the presence of metabolic risk factors such as obesity, diabetes, insulin resistance, or evidence of metabolic dysregulation. This change aims to better capture the complex interactions between liver fat accumulation and metabolic health [84]. Similar to NAFLD, MAFLD is mostly observed in obese patients, but it can also occur in lean individuals. A recent study revealed a novel circulating histone signature by using a rapid and noninvasive imaging technology called ImageStream (X), which has the ability to differentiate the severity of steatosis in individuals with lean MAFLD [85]. This assay can be used for the detection of potential human lean MAFLD markers, via the analysis of intact histones and histone complexes, which are released into the blood circulation as “liquid biopsies” from dying cells [85]. In particular, a significant decrease in the expression of histone variants macroH2A1.1 and macroH2A1.2 was observed, either individually or as a natural dimer with H2B [85] (Figure 2). Notably, the downregulation of macroH2A1.2 was nearly twice as significant as that of macroH2A1.1 in lean patients with steatosis grade 3 compared with those with grade 1 [85] (Figure 2). In contrast, the expression of these histones did not change significantly in the overweight subgroup [85].

A recent study revealed that changes in the macrophage epigenome of lean MAFLD patients can repress the bile acid-associated signaling pathway and the anti-inflammatory responses downstream [86] (Figure 2). A metabolic–epigenetic axis governs both the inflammatory and metabolic responses in macrophages, as well as the production of proinflammatory cytokines. Metabolic endotoxemia has been shown to have a widespread impact on the structure of chromatin [87]. Transcriptome analysis and epigenomic reprogramming analysis showed that, in lean MAFLD patients, metabolic endotoxemia induces the activation of proinflammatory genes and the secretion of Toll-like receptor 4 (TLR4), which hampers bile acid signaling [86]. These events culminate in an augmented production and secretion of bile acids accompanied by suppressed bile acid signaling, which aligns with the loss of metabolic adaptation and the progression of liver disease [86] (Figure 2).

## 5. Noncoding RNAs

Noncoding RNAs (ncRNAs) constitute a substantial part of the transcriptome and lack discernible protein-coding functions. However, ncRNAs have been involved in a wide range of biological processes, including disease pathogenesis [88]. Advancements in sequencing technology and data analysis have allowed researchers to discover numerous ncRNAs, including long noncoding RNAs (lncRNAs) [89], circular RNAs (circRNAs) [90], and small ncRNAs [91]. miRNAs, a subgroup of small ncRNAs, are endogenous single-stranded RNAs that play a crucial role in the regulation of biological processes and epigenetic mechanisms [92,93] (Figure 3).

The dysregulation of miRNA expression profiles has been implicated in the pathophysiology of various diseases, and distinct miRNA expression profiles have been identified to be associated with NAFLD [94,95]. Particularly, miRNAs have emerged as reliable circulating biomarkers for the noninvasive diagnosis and assessment of NAFLD severity [96].

miRNA miR-122, which exhibits high expression levels in the human liver, has been found to accelerate the progression of NAFLD [97]. Conversely, both miR-122 and miR-223 have shown potential in ameliorating NAFLD [98]. Additionally, a study by Liu et al. suggested that the expression of miR-192 contributes to the progression of NAFLD [99]. These miRNAs have also been proposed as diagnostic biomarkers for liver injury and potential therapeutic targets [100]. However, limited research has explored the role of miRNAs in detecting lean NAFLD in the absence of obesity.

A recent study investigated the role of serum miRNAs in lean NAFLD and their potential as biomarkers [101]. Serum miR-4488 expression levels were found to be augmented in lean NAFLD patients compared with obese NAFLD patients and healthy controls [101] (Figure 4).

Gene Ontology (GO) and Kyoto Encyclopedia Genes and Genomes (KEGG) enrichment analyses were used to identify the miR-4488 target gene prediction and pathway analysis [101]. The choline metabolism in tumorigenesis, the signaling pathway of tumor necrosis factor (TNF), and the p53 signaling pathway were found enriched, suggesting that miR-4488 may affect the lean NAFLD progression by taking part in these signaling pathways [101] (Figure 4). In parallel, various genes such as ARHGAP1, SLC10A1, SIX5, CTNNA1, and WTIP were identified as miR-4488 regulatory targets associated with lean NAFLD [101] (Figure 4).

The circulating levels of miR-122, an essential factor for glucose and lipid metabolism [102], and miR-33a/b*, a main modulator of fatty acid and cholesterol homeostasis [103], were investigated in relation to their hepatic expression in women with NAFLD [104]. The expression of miR-33b* in the liver was related to the presence of obesity, as it was increased in morbidly obese compared with moderately obese and normal-weight NAFLD women [104] (Figure 4). In contrast, the hepatic expression of miR-122 was reduced in the morbidly obese cohort compared with the moderately obese subpopulation [104]. In this regard, previous research has indicated that various miRNAs play a role in the regulation of adiposity and insulin sensitivity. A positive association between increased levels of circulating miR-122 and both obesity and insulin resistance in young adults has been demonstrated [105]. Additionally, there is evidence of dysregulation of circulating miRNAs in cases of severe obesity, with significant alterations in this miRNA profile occurring as a result of weight loss induced by bariatric surgery [106].

Vonhögen et al. proposed the use of miR-216a as a biomarker for obesity and its associated metabolic diseases in women [107]. In this study, the miR-216a gene was identified as an obesity-susceptibility locus, containing CpG islands that exhibit varying levels of DNA methylation between obese and nonobese children. Moreover, the DNA methylation patterns at this locus were correlated with distinct circulating miR-216a plasma levels in both obese and nonobese women [107] (Figure 4).

The characterization of the intestinal microbiome in NAFLD patients revealed an increase in the bacterium Escherichia Shigella; this increase was closely related to NAFLD severity independently of obesity [108]. However, a study in rats showed that the species E fergusonii promoted the development of lean NAFLD, as nonobese animals demonstrated hepatic steatosis and hepatocyte ballooning [108] (Figure 4). The presence of E fergusonii impaired the host lipid metabolism via the suppression of lipid β-oxidation in the liver and the induction of de novo lipogenesis [108] (Figure 4). In deep sequencing analysis, E fergusonii-derived microRNA-sized, small RNA (msRNA) 23,487 was found to reduce the expression of host hepatic peroxisome proliferator-activated receptor α (PPARa) [108]. Through this process, the aggregation of lipids in the liver may be promoted via the secretion of msRNA 23487, potentially contributing to steatohepatitis and the fibrosis pathogenesis of lean NAFLD in rats [108] (Figure 4).

NAFLD-associated fibrosis and sarcopenia share common pathophysiological pathways, including chronic inflammation, insulin resistance, and changes in the modulation of various proteins and hormones, which possibly explain the bidirectional influence between these disorders [109]. In Western NAFLD patients, sarcopenia has been related to the severity of liver fibrosis and steatosis [110]. The prevalence of skeletal muscle loss is frequently observed in lean NAFLD patients, whereas the frequency of sarcopenic obesity is rare in NAFLD patients [111]. Regular exercise and diet modification are effective strategies in alleviating steatosis in lean NAFLD [112]. A recent study has shown that lifelong regular exercise in rats improved skeletal muscle atrophy, impaired autophagy, mitochondria dysfunction, and apoptosis. This amelioration in skeletal muscle aging may be the result of the augmented miR-486 expression and the following activation of the PI3K/Akt pathway and the indirect suppression of HIPPO signaling [113] (Figure 4).

Cell-to-cell communication is mediated by extracellular vesicles (EVs), which have been used as biomarkers and drug carriers [114,115]. Microvesicles, apoptotic bodies, and exosomes constitute the main types of EVs; exosomes are membrane-bound EVs that contribute to antigen presentation, intercellular communication, and mRNA and miRNA shuttling [116]. MiRNAs can be enclosed within exosomes and released from cells. Ying et al. showed that in obese mice adipose tissue macrophages (ATMs) released exosomes containing miRNAs (Exos) [117]. Exos administration in lean mice caused glucose intolerance and insulin resistance [117]. Conversely, the administration of ATM Exos of lean mice in obese recipients critically improved glucose tolerance and insulin sensitivity [117]. One of the specific miRNAs found in abundance in obese ATM Exos was miR-155, known to target PPARγ, suggesting that this miRNA may be the causal factor for insulin resistance induction [117].

Translational research in the field of miRNAs holds promise for the development of novel diagnostic and therapeutic strategies that aim to treat lean NAFLD, considering the critical regulatory role of miRNAs for this condition. This avenue of research may lead to innovative approaches that can effectively diagnose and treat lean NAFLD by targeting miRNA-related mechanisms.

### ncRNA Differences between Obese and Lean NAFLD

When comparing obese and lean individuals with NAFLD, studies have focused on differences in the expression and function of various types of ncRNAs. ncRNA-mediated epigenetic alterations influence a variety of metabolic pathways and cellular activities in the hepatic tissue, including oxidative stress, inflammatory immune response, hepatic glucose and lipid metabolism, and even tumorigenesis [118]. The differential expression of miRNAs has been observed between obese and lean individuals with NAFLD. These miRNAs can target the genes involved in lipid metabolism, inflammation, fibrosis, and other pathways relevant to NAFLD progression [119,120,121]. CircRNAs are a type of ncRNA that form closed-loop structures due to a covalent bond between 3′ and 5′ ends [122]. They can act as microRNA sponges or regulate protein function. Altered circRNA expression profiles have been associated with NAFLD and obesity, affecting the pathways involved in insulin resistance and lipid metabolism [123]. Lastly, several lncRNAs have been found to be dysregulated in NAFLD, potentially influencing the pathways related to inflammation, lipid metabolism, and fibrosis [118].

## 6. Conclusions and Prospects

The underlying mechanisms of NAFLD are multifaceted; both genetic and nongenetic factors are essential for the initiation and development of this disorder. Lean or nonobese NAFLD is characterized as an even more complex and multifactorial disorder. While certain risk factors like patients’ age and genetic code are unchangeable, other risk factors can be modified during life depending on lifestyle alterations and medical interventions. Understanding the epigenetic regulation of lean NAFLD is crucial for the identification of potential therapeutic targets and the development of personalized treatment strategies. In future research, precision medicine based on gene alterations, including the use of RNA interference and anti-sense oligonucleotides, could potentially modify the influence of genes on the pathogenesis and severity of NAFLD, making genetics a modifiable factor. Future studies on the role of genetics and epigenetics will enhance our understanding of the underlying mechanisms of lean or nonobese NAFLD, thereby enabling the development of potential therapeutic strategies. Acquiring a better understanding of modifiable risk factors would also facilitate the prevention or delay of NAFLD progression.

Additionally, therapeutic strategies encompass the potential regulation of the enzymes responsible for epigenetic modifications of DNA and proteins. In parallel, the regulation of gene expression is influenced by miRNAs. Assessing circulating miRNA profiles holds promise as a noninvasive method to evaluate and monitor the severity of liver diseases. However, independent validations are necessary to further establish the reliability and effectiveness of this approach. Lastly, future research in genomic regions with DMRs may highlight them as novel biomarkers for the progression of carcinogenesis or as a therapeutic approach, which indicates the significance of studying epigenetic alterations.

Although there is much to be learned about the influence of epigenetic modifications on the mechanisms contributing to NAFLD pathogenesis in humans, efforts to gain knowledge on the epigenetic-based diagnostic and therapeutic potency and evaluation tools would have a beneficial impact on precision medicine in metabolic diseases and overall human health. Incorporating molecular diagnosis into the clinical assessment of individuals with lean NAFLD has the potential to yield precise diagnostic information, allowing for targeted therapeutics.

## Figures and Tables

**Figure 1 ijms-24-12864-f001:**
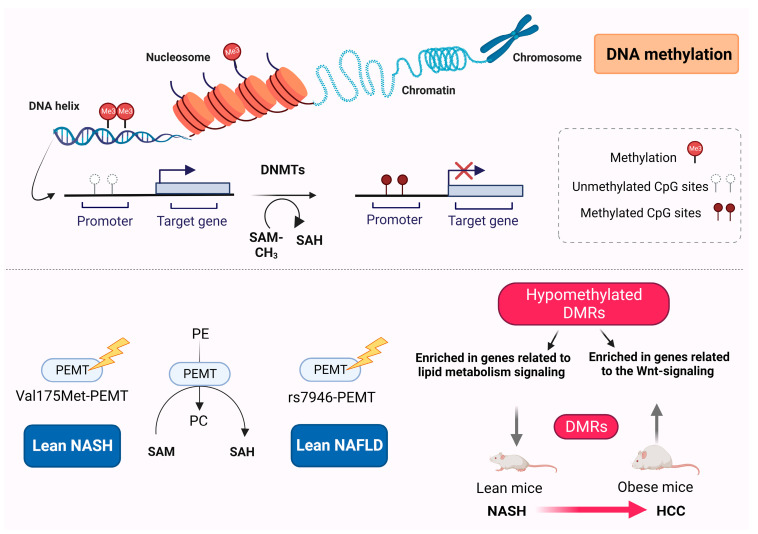
DNA methylation in lean NAFLD. DNA methylation is a post-translational modification through which methyl groups (Me) are added to DNA on the CpG islands and regulates transcriptional gene expression, particularly gene silencing. The rs7946-PEMT genetic variant in the PEMT gene has been related to increased risk for lean NAFLD, whereas Val175Met-PEMT has increased incidence in lean NASH patients. Differential methylation regions and differentially methylated genes have been observed in both lean and obese mice during the progression of nonalcoholic steatohepatitis to hepatocellular carcinoma. This figure was generated using BioRender, available online at: https://biorender.com (accessed on 25 July 2023). Abbreviations: DNMTs, DNA methyltransferases; SAM, S-adenosylmethionine; SAH, S-adenosylhomocysteine; PEMT, phosphatidylethanolamine N-methyltransferase; NASH, nonalcoholic steatohepatitis; NAFLD, nonalcoholic fatty liver disease; PE, phosphatidylethanolamine; PC, phosphatidylcholine; DMRs, differentially methylated regions; HCC, hepatocellular carcinoma.

**Figure 2 ijms-24-12864-f002:**
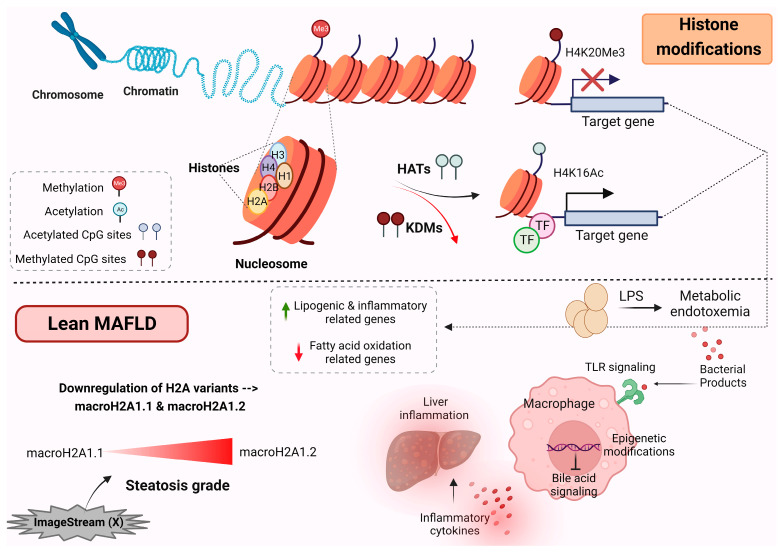
Modifications of histones. A major association between chromatin modifications and metabolic imbalance has been observed in lean individuals with MAFLD. Methylation and acetylation of histones have resulted in the stimulation of genes closely related to lipogenic and inflammatory processes and the downregulation of genes associated with fatty acid oxidation. In lean MAFLD, a significant reduction in macroH2A1.1 and macroH2A1.2 histone variants has been observed. Macrophage epigenome modifications in lean MAFLD downregulate the bile acid signaling and induce an inflammatory response, leading to more severe liver disease. This figure was generated using BioRender, available online at: https://biorender.com (accessed on 25 July 2023). Abbreviations: HATs, histone acetyltransferases; KDMs, histone lysine demethylases; TF, transcription factor; MAFLD, metabolic-associated fatty liver disease; TLR, Toll-like receptor; LPS, lipopolysaccharide.

**Figure 3 ijms-24-12864-f003:**
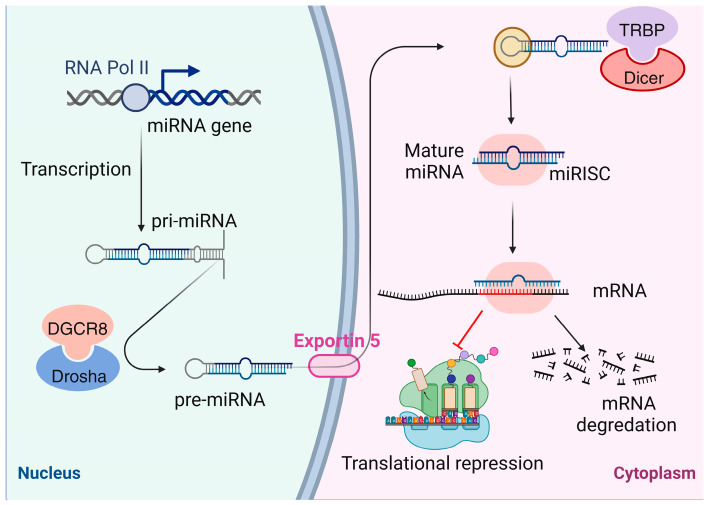
miRNA structure and biogenesis. MicroRNA (miRNA) genes, a subgroup of small noncoding RNAs, are typically transcribed by an RNA polymerase II within the nucleus to generate pri-miRNA transcripts. The microprocessor complex, comprising the dimeric RNA-binding protein DGCR8 and the RNase III enzyme Drosha, facilitates the cleavage of pri-miRNA, producing the pre-miRNA precursor. Subsequently, exportin 5 (XPO5) transports the pre-miRNA to the cytoplasm. In the cytoplasm, the pre-miRNA is cleaved into a mature double-stranded miRNA by the Dicer enzyme and the transactivation response element RNA-binding protein (TRBP) complex. Upon maturation, the mature miRNA becomes part of the miRNA-associated multiprotein RNA-induced silencing complex (mi-RISC). Following that, the mature miRNA binds to complementary regions on the target mRNA, guiding and modulating its expression through base pairing. Generally, mature miRNA binds to specific mRNA 3′-untranslated sequences (3′-UTR) via partially complementary regions, leading to the inhibition of mRNA translation into proteins. However, if miRNA and mRNA exhibit high complementarity, this can result in the cleavage of the target mRNA. This figure was generated using BioRender, available online at: https://biorender.com (accessed on 25 July 2023). Abbreviations: miRNA, microRNA; pol II, polymerase II; DGCR8, DGCR8 microprocessor complex subunit; miRISC, miRNA-induced silencing complex; TRBP, transactivation response element RNA-binding protein.

**Figure 4 ijms-24-12864-f004:**
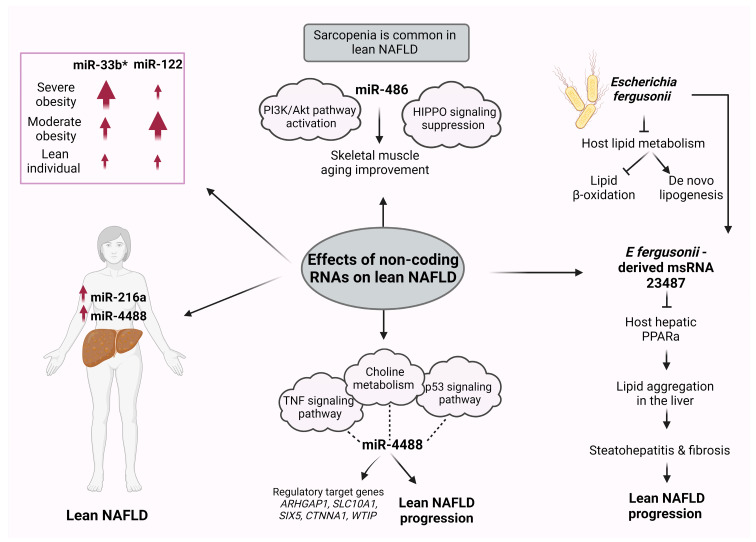
Role of miRNAs in lean NAFLD. Several miRNAs have been implicated in the pathogenesis of lean NAFLD and have been suggested as potential biomarkers for lean NAFLD diagnosis and prognosis. This figure was generated using BioRender, available online at: https://biorender.com (accessed on 25 July 2023). Abbreviations: miRNA, microRNA; NAFLD, nonalcoholic fatty liver disease; TNF, tumor necrosis factor; p53, protein P53; PPARa, peroxisome proliferator-activated receptor alpha; msRNA, microRNA-sized small RNA.

## Data Availability

Not applicable.

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
