# Peer review of "Epigenetic Regulation in Lean Nonalcoholic Fatty Liver Disease"

_ijms, 2023, doi:10.3390/ijms241612864_

Round 1

Reviewer 1 Report

Dear  Dr. Triantos and  coauthors,

this is very well written  review of high relevance, which might find  broad interest. I have only minor concerns:

1) In the review, you highlight the histone changes and miRNA expression pattern in lean NAFLD.  In these sections, could you explain what are the similarities with the changes in obese NAFLD and what are the differences?

2)  In my opinion, Figure 3 is a bit overloaded.   I think it would be good to take out some information and then structure the figure a bit more clearly.  I would either take out the "non-coding RNA" part and describe its function in the text only, or I would take out the miRNA synthesis and function shown in the square in the middle of the figure. 

Author Response

Reviewer 1

Dear Dr. Triantos and coauthors,

this is very well written review of high relevance, which might find broad interest. I have only minor concerns:

Comment 1: In the review, you highlight the histone changes and miRNA expression pattern in lean NAFLD.  In these sections, could you explain what are the similarities with the changes in obese NAFLD and what are the differences?

Response to comment 1: In the revised manuscript we have added a paragraph at the end of DNA methylation section (page 5, lines 203-216) and non-coding RNAs section (page 11, lines 436-450), briefly discussing the differences observed in epigenetic regulation between lean and obese NAFLD individuals, as suggested by reviewer.

Comment 2: In my opinion, Figure 3 is a bit overloaded. I think it would be good to take out some information and then structure the figure a bit more clearly.  I would either take out the "non-coding RNA" part and describe its function in the text only, or I would take out the miRNA synthesis and function shown in the square in the middle of the figure.

Response to comment 2: We have revised the figure 3. In the revised manuscript, the biogenesis and structure of miRNA is reported as figure 3, whereas the role of miRNAs in lean NAFLD is a separate figure and is reported as figure 4, in order to be clearer and more specific, as reviewer suggested.

Reviewer 2 Report

only minor editing required

Author Response

Reviewer 2

The Review Article IJMS 2560912 entitled " Epigenetic regulation in lean non-alcoholic fatty liver disease” by Ioanna Aggeletopoulou, Maria Kalafateli, Efthymios P Tsounis and Christos Triantos describes different epigenetics factors such as methylation, histone modification and microRNA regulation that can be behind the influencing the development of lean NAFLD. Furthermore, the authors highlight the therapeutic implications of these modifications and their potential use as biomarkers for the disease.

Comment 1: The manuscript is well organized as well as well written with a nice introduction and conclusion and will be of interest for the scientific community after minor revisions. In general, each section is clear and well addressed, however I feel section 2 regarding lean NAFLD subtypes does not make much sense as they are not referred again throughout the whole text. I would suggest either further comments on this aspect in relationship to the different epigenetic mechanisms described or just introduce lean NAFLD subtype into the introduction section.

Response to comment 1: As there is no related references on the differences between the epigenetic modifications on subtype 1 and subtype 2 of lean NAFLD, this section has been merged with the Introduction Section, as suggested by the reviewer (page 2, lines 62-72).

Comment 2: The second comment has to do with the use of NAFLD and MAFLD acronyms. The authors always refer to NAFLD, however in section 5 analyzing histone modification they do not refer to NAFLD but to MAFLD. It would be worth to clarify both concepts as it is not clear enough whether they can use indistinctly.

Response to comment 2: We are in totally agreement with the comment of reviewer. Although the MAFLD is different entity of NAFLD, these disorders share common features. Thus, we thought it would be interesting to incorporate this information into our Review, as the literature on the epigenetic regulation in lean NAFLD and lean MAFLD is still limited. In the revised manuscript, we have tried out to clarify and make it clear that we are talking about two different entities, as suggested by the reviewer (page 7, lines 273-282). 

Comment 3: Additionally, the whole text should be carefully revised to prevent the last word of each line from being split in an incorrect manner.

Response to comment 3: We have revised as suggested.  

Comment 4: Comments on the Quality of English Language - only minor editing required.

Response to comment 4: We have edited the manuscript for correct English language.